# Chronic Effects of Different Intensities of Interval Training on Hemodynamic, Autonomic and Cardiorespiratory Variables of Physically Active Elderly People

**DOI:** 10.3390/ijerph20095619

**Published:** 2023-04-24

**Authors:** Leandro Sant’Ana, Diogo Monteiro, Henning Budde, Aline Aparecida de Souza Ribeiro, João Guilherme Vieira, Estêvão Rios Monteiro, Fabiana Rodrigues Scartoni, Sérgio Machado, Jeferson Macedo Vianna

**Affiliations:** 1Post Graduate Program in Physical Education, Federal University of Juiz de Fora, Juiz de Fora 36036-900, MG, Brazil; 2Strength Training Studies and Research Laboratory, Federal University of Juiz de Fora, Juiz de Fora 36036-900, MG, Brazil; 3ESECS, Polytechnic of Leiria, 2411-901 Leiria, Portugal; 4Life Quality Research Centre (CIEQV), 2040-413 Leiria, Portugal; 5Research Center in Sport, Health, and Human Development (CIDESD), 5001-801 Vila Real, Portugal; 6Institute for Systems Medicine (ISM), Faculty of Human Sciences, Medical School Hamburg, University of Applied Science and Medical University, 20457 Hamburg, Germany; 7Post Graduate Program in Physical Education, Federal University of Rio de Janeiro, Rio de Janeiro 21941-599, RJ, Brazil; 8Post Graduate Program in Rehabilitation Sciences, Augusto Motta University Center, Rio de Janeiro 20911-300, RJ, Brazil; 9Graduate Program in Physical Education, IBMR University Center, Rio de Janeiro 22631-002, RJ, Brazil; 10Sport and Exercise Science Laboratory, Catholic University of Petrópolis, Petrópolis 25685-100, RJ, Brazil; 11Departament of Sports Methods and Techniques, Federal University of Santa Maria, Santa Maria 97105-900, RS, Brazil; 12Laboratory of Physical Activity Neuroscience, Neurodiversity Institute, Queimados 26325-020, RJ, Brazil

**Keywords:** interval training, cardioprotection, elderly, hemodynamic variables, heart rate variability, maximal oxygen consumption

## Abstract

Interval training (IT) is a very efficient method. We aimed to verify the chronic effects of IT with different intensities on hemodynamic, autonomic and cardiorespiratory variables in the elderly. Twenty-four physically active elderly men participated in the study and were randomized into three groups: Training Group A (TG_A_, *n* = 8), Training Group B (TG_B_, *n* = 8) and control group (CG, *n* = 8). The TG_A_ and TG_B_ groups performed 32 sessions (48 h interval). TG_A_ presented 4 min (55 to 60% of HRmax) and 1 min (70 to 75% of HRmax). The TGB training groups performed the same protocol, but performed 4 min at 45 to 50% HRmax and 1 min at 60 to 65% HRmax. Both training groups performed each set six times, totaling 30 min per session. Assessments were performed pre (baseline) after the 16th and 32nd intervention session. The CG performed only assessments. Hemodynamic, autonomic and cardiorespiratory (estimated VO_2max_) variables were evaluated. There were no significant differences between protocols and times (*p* > 0.05). However, the effect size and percentage delta indicated positive clinical outcomes, indicating favorable responses of IT. IT may be a strategy to improve hemodynamic, autonomic and cardiorespiratory behavior in healthy elderly people.

## 1. Introduction

According to the World Health Organization, the number of older people has increased significantly [1]. Aging is natural and inevitable, so it will be in constant progress in the coming years across the world [2]. Because of this, the American College of Sports Medicine (ACSM) and the American Heart Association (AHA) directed several positions on the importance of physical training in improving cardiovascular and cardiorespiratory fitness, specifically for the elderly population [3]. Aging brings several changes in the cardiovascular system [4]. Physiological changes in aging can be partially reduced by regular physical exercise to improve the cardiovascular system [5,6] and cardiorespiratory fitness [7]. Disorders related to the cardiovascular and cardiorespiratory systems are the leading cause of disease and mortality worldwide [8,9] and appear to have more significant effects on the elderly [10]. Additionally, with the aging process, the cardiovascular functional decline is significant [11], where the maintenance and improvement of this system are essential for the organic integrity of the elderly [12].

The cardiovascular system, among other factors such as endocrine factors, is driven by autonomic [13] and hemodynamic [14] actions, in which an integrated way provides the functional efficiency of this system [15]. The cardiorespiratory system is also essential in cardiovascular potential and efficiency, which can suffer significant functional reductions due to aging [16]. However, these systems require attention in the conditional improvement of their actions [17]. Studies report that the harmful physiological effects in the elderly are not only related to aging but also lifestyle habits [18], such as regular physical activity [19], and that, especially for the autonomic system, aging is not a limiting factor [20]. Therefore, the improvement in physical conditioning is linked to cardiovascular and cardiorespiratory efficiency, especially in the face of aging [21,22].

To maintain and improve cardiovascular and cardiorespiratory functions in older men, one of the strategies is to promote an increase in aerobic capacity, including for example, maximum oxygen uptake (VO_2max_) [23]. Towards this objective, interval training (IT) has excellent potential for cardiovascular and cardiorespiratory improvement [24]. Previous studies have reported increases in VO_2max_ to reduce the risk of death from cardiovascular and cardiorespiratory events (±15%) [8]. Therefore, to improve the aerobic capacity, specific training is required, and IT seems to promote chronic response [25]. Additionally, IT is suggested to improve the maximum aerobic profile in the elderly [26] and increase autonomic and hemodynamic balance [27], thus consolidating an improvement within the cardiovascular system [28].

IT-related studies have been applied using stimulus intensities above metabolic and ventilatory thresholds, such as training with high-intensity intervals. Additionally, studies have used IT to assess individuals of different ages and objectives for physiological analyses [29]. Therefore, currently, there is little research using IT at intensities below the physiological thresholds, including for example, anaerobic and ventilatory threshold [28], especially for the elderly. In IT, stimuli performed below physiological thresholds (e.g., anaerobic and ventilatory 2) can provide less mechanical and physiological wear in the elderly. Further, uptime can be shorter and more motivating, which would be interesting for this audience [30].

In the elderly, another factor that can hinder the practice of some activities at higher intensities is the motor pattern. In IT, the time at higher intensities is shorter, which may provide greater fitness for the practice of exercise, and with that, IT can be a suitable training prescription option [11,31]. However, conditional improvements in this population become indispensable since the physical fitness index [10] is a determining factor in enabling more remarkable preservation of functional and organic efficiencies and, thus, minimizing the deleterious effects caused by the low physical fitness added to the aging process [19]. Additionally, gains in body balance levels will also be of paramount importance in avoiding the risk of falls [10,11,12,13,14,15,16,17,18,19,20,21,22,23,24,25,26,27,28,29,30,31,32].

### Present Study

Research on the elderly and cardioprotective variables is extremely important, mainly due to the fact that the aging process directly affects the systems that generate cardioprotection [10,11,33]. The elderly population is growing significantly [2] and studies about this are important for us to know more about these individuals, especially in relation to the practice of physical exercise [34]. It seems that IT is a great option for training programs with the elderly, especially on the variables related to cardioprotection [24]; however, despite this, there are still only a few studies on the intervention with IT investigating cardioprotective variables of the elderly [18,35,36]. Because of this need, the objective of this study is to verify the chronic effects of IT with different intensities on hemodynamic, autonomic and cardiorespiratory variables of physically active older people. Our hypothesis was that somehow, protocols with different intensities of IT could offer (positive) changes in the dependent variables researched here, with these being hemodynamic [37], autonomic [18] and cardiorespiratory [24].

## 2. Materials and Methods

### 2.1. Participants

For the selection of the participants, the simple random sampling method was adopted. Thus, the sample consisted of 24 elderly men (age: 68.8 ± 6.8 years; body mass: 74.4 ± 18.1 kg; height: 1.70 ± 0.8 m; BMI: 25.1 ± 2) physically active (Table 1). As for the selection of participants, the following conditions were considered: older men (aged 60 or over) who are regular practitioners of physical activity, but with relatively low energy expenditure and below the minimum activity time per week (150 min), 2 to 3 times a week, 1 h per session.

These activities should not be related to aerobic training with the same characteristics to be applied in this study, and everyone should have competent physical conditions to perform the intervention proposed by the study. On the other hand, the exclusion criteria were any pharmacological medication and ergogenic resources (blood pressure control drugs, beta-blockers, among others related to cardiovascular and cardiorespiratory control) that could influence the expected results in some way presenting musculoskeletal disorders that compromise training. In addition, all participants received a recommendation not to eat foods that could interfere with cardiovascular and cardiorespiratory responses (excessive consumption of salt, caffeine, alcohol, and high-calorie foods, among others). After explaining the risks and benefits of the research, the participants agreed to participate and completed the Physical Activity Readiness Questionnaire (PAR-Q) and signed an informed consent form following the Declaration of Helsinki, which complies with Resolution 466/12 of the National Health Council (CNS).

### 2.2. Procedures

Participants were randomized to the groups by one of the researchers (no participation in the training sessions: Machado S.) using the draw method, as follows: Once the 24 individuals were selected, all were entered for a selection of 16 participants for the training groups. After 16 individuals were drawn into the training groups, the others (*n* = 8) were directly allocated to the control group (CG). The 16 elderly people selected for the interventions were again drawn 4 by 4 (separated by blocks, totaling 4). In the first block, the first two drawn were allocated to Training Group A (TG_A_) and the last two to Training Group B (TG_B_). In the second block, the first two drawn were entered into the TG_B_ and the last two into the TG_A_. In the third block, the first and fourth drawn went to the TG_A_, while the second and third went to the TG_B_. Finally, in the fourth block the first and third draw were directed to the TG_A_ and the second and fourth to the TG_B_. After this procedure, the participants formed the three groups of the present research: the TG_A_ group (*n* = 8), the TG_B_ (*n* = 8) and the CG (*n* = 8) [38]. For groups that trained, the interventions were developed for 32 sessions, with a 48 h interval between one and the other session. One of the participants in the TG_B_ group had suspended their participation in the research (at the beginning of the interventions) for personal reasons. Thus, other individuals were selected to compose the group, maintaining the same number of individuals in the three groups (*n* = 8). There was no dropout in the TG_A_ and CG (Figure 1). All evaluations (hemodynamic, autonomic and cardiorespiratory) were performed in the pre (baseline) moments after the 16th and 32nd sessions of the invention, including hemodynamic (heart rate and blood pressure), autonomic (heart rate variability) and cardiorespiratory (estimated VO_2max_). The control group did not carry out any intervention; they continued with their daily domestic activities without carrying out activities such as walking, cycling or any other activities that could interfere with the level of physical fitness already existing in these individuals. However, they carried out the evaluations during the same period as the training group.

### 2.3. Measures

#### 2.3.1. Analyses of Hemodynamic and Autonomic Variables

Blood pressure (BP) values were collected in the left arm [39]. For hemodynamic analysis, the values of resting heart rate (HRR), systolic blood pressure (SBP), and diastolic blood pressure (DBP) were analyzed. The variables were analyzed within 10 min, with the subjects resting in the sitting position. For HRR, SBP and DBP, the average of the 8th and 9th minutes of measurement performed in 10 min was used. After the BP data was determined, the mean arterial pressure (MBP) was calculated using MBP = SBP + (DBP × 2)**/**3. Then, using the HRR, the double product (DP) was calculated using the equation: [HRR (bpm) × SBP (mmHg)]. The DP represents the workload or oxygen demand of the heart and is considered a non-invasive reference for cardiac overload [40].

For the autonomic analysis, the behavior of heart rate variability (HRV) was used, measured in a 5 min window within the rest period. Considering the mean and indices of the time domain (RR, RMSSD and SDNN) and frequency (LF, HF and LF**/**HF), both were calculated by software specific to this type of treatment (Kubios HRV Standart, 3.3.1). Time-domain: normal RR (time between two adjacent heartbeats) was calculated and, after that, based on statistical or geometric methods (mean, standard deviation, and indexes derived from the RR intervals of the histogram), the indices of fluctuations in the duration of cardiac cycles were calculated, which are the RMSSD (square root of the mean of the square of the successive differences between the adjacent normal RR intervals, in a time interval, expressed in ms) and the SDNN (standard deviation of all the normal RR intervals recorded in a time interval, shown in ms). The RMSSD represents parasympathetic activity, and SDNN represents sympathetic and parasympathetic activity, but it does not allow us to distinguish when the HRV changes are due to the increase in sympathetic tone or the withdrawal of vagal tone [41].

For the analysis of HRV in the frequency domain, low-frequency components (low frequency—LF) were used, which corresponds to the joint action of the parasympathetic and sympathetic on the heart with the predominance of the sympathetic and the high-frequency component (high frequency—HF) that corresponds to respiratory modulation and represents the activation of the vagus nerve [42]. Finally, we use the LF**/**HF ratio that we call sympathetic-vagal balance. Regarding the LF**/**HF ratio as an indicator of sympathetic-vagal balance, there are still controversies [43] due to the veracity of this index as a parameter of autonomic balance [44].

A POLAR RS800CX watch (Multisport model), Kempele, Finland**^®^** [45] was used to collect HRR and HRV. For the analysis of BP, a digital oscillometric device of the brand OMRON M6 (HEM-7001- E)**^®^** was used [46]. For HRV treatment, the data was transferred to the computer and attached to the Polar Trainer 5 Software**^®^**. Correction procedures for all data were carried out on this platform and were subsequently filed in TXT format for the start of treatment in the Kubios HRV Standard Software (using 300 s inter-beat intervals), version 3.3.1. All the collected data is calculated and presented in different standards to have broad interpretations of HRV.

#### 2.3.2. Analyses of Estimated Oxygen Consumption

The maximum oxygen consumption (VO_2max_) was estimated using the indirect method [47], which was evaluated using the following protocol: 3 min at 5.0 km/h with 1% inclination. From this initial stage, increments of 2% on the slope (approx. 1 MET) were administered every minute aiming to reach the minimum intensity of 65% of HRRes. Once reached, the incline and speed were then kept unchanged for 6 min to allow the steady state to be reached [48]. At the end of this phase, it was expected that the HR would be stabilized at approximately 70% of HR_res_. The criteria for interruption for both tests followed the recommendations of the ACSM [49]. VO_2max_ was obtained by the walking equation, VO_2max_ = [0.1 (speed) + 1.8 (speed) (incline/100) + 3.5], in which speed is given in m/min. Finally, the VO_2max_ was predicted by the equation: VO_2max_ = [(VO_2max_ − 3.5)**/**% HR_res_ + 3.5], in which VO_2max_ is expressed in mL·kg**^−^**^1^·min**^−^**^1^ [49]. The test was carried out on a treadmill under the Movement brand (model RT 250) **^®^**.

### 2.4. Training Protocols

The interval training protocol was performed in the same format for both experimental groups but with different intensities based on the study by Pichot et al. [18]. Due to the participants’ motor restrictions and for safety reasons, it was determined that the interventions were carried out by walking (with an incline increase, when necessary, at the time of the stimulus) on a treadmill. For interval training, a 1:4 ratio setting was applied. Intensities were controlled by calculations based on the maximum heart rate (HR_max_) [50], adjusted with the reserve heart rate [51]. The TG_A_ performed 4 min with intensity relative to 55 to 60% of reserve heart rate (HR_res_) and 1 min to 70 to 75% of HR_res_. The TG_B_ training group, in turn, performed the same protocol but performed 4 min at 45 to 50% of HR_res_ and 1 min at 60 to 65% HR_res_. Each four-by-one sequence is considered a block in both training groups (A and B), accounting for 6 blocks. The equivalent is thus thirty minutes in duration. At the end of each block, the perception of effort was measured [52] to help with the proposed intensity control. All training sessions were carried out on a treadmill using the Movement brand (model RT 250)^®^. All participants were trained by only one researcher (Sant’Ana, L.).

### 2.5. Statistical Analysis

A sample calculation was performed (G*power) [53], anticipating a “large” effect size (f = 0.4), with an α = 0.05, a statistical power of (1 − β) = 0.95, the correlated dependent variables with an r = 0.50, and a violation of sphericity (ε) = 0.80, will require a total sample size of 21 individuals and an enabled power of 0.95. The suggested effect size and the remaining parameters were defined according to similar studies that evaluated changes in cardioprotective variables during exercise protocols of the elderly [18,24,37].

In the descriptive analysis, the means and standard deviation of the variables were calculated. Normality was not rejected by the Shapiro–Wilk test and by the histogram and Q-Q Plot analysis, and homoscedasticity was confirmed by the Mauchly test, which suggests a normal distribution for the collected data implying the possibility of parametric inferential treatment. The analysis of variance (ANOVA mixed effects model) with repeated measures was applied to test the main and interaction effects. Additionally, following the recommendations of the ACSM [54], for a more clinical determination of the acquired results [55], the method of effect size (ES) analysis was applied [56]. Further, the percentage delta (∆%) calculation was also used. The ES was calculated using the formula d = Md/Sd, where Md is the mean difference, and SD is the standard deviation of the differences. The ES was defined as small (≥0.2), medium (≥0.5) and large (≥0.8) [56]. The percentage delta of a variable is calculated using the final (FV) and initial (IV) value, where the formula: ∆% = (FV/IV − 1) × 100. All statistical analyses were performed using GraphPrism software version 8.0.1, with a significance level of 5% (*p* < 0.05). For the sample calculation, GPower 3.1 software was used [53,54,55,56,57].

## 3. Results

Twenty-four older people were selected (Table 1), and they were randomly divided into TG_A_, TG_B_ and CG, with 8 participants for each group.

The effect size, ***p***-values, and Δ% for each condition were presented in Table 2. There were no differences between baseline values. There were no significant (*p* > 0.05) differences within and between protocols at any time point for resting heart rate, blood pressure, double product, VO_2max_ estimate (mL·kg^−1^ min^−1^) (Table 2) and heart rate variability (Table 3). The ES indicated that all experimental protocols showed improvements in the variables blood pressure with variation between −0.86 (Large) and −1.11 (Large), rate pressure product with −0.52 (Medium), heart rate variability with variation between −0.49 (Medium) and −4.00 (Large), VO_2max_ estimate (mL·kg^−1^·min^−1^) with variation between 0.82 (Large) and 1.12 (Large) when compared to control group.

## 4. Discussion

According to global positions [58], dysfunctions in the variables of cardioprotective function (HR and BP) are the leading causes of illness and death in the world, especially in the elderly [59]. Thus, this study aimed to verify the chronic effects of IT (using walking mechanics) on cardiovascular and cardiorespiratory variables of physically active older people. There were hemodynamic (HR_R_, SBP, DBP, MBP and DP), autonomic (HRV), and cardiorespiratory variables (VO_2max_ estimate—mL·kg ^−1^·min^−1^). The intervention conducted was composed of 32 interval training sessions for two training groups (TG_A_ and TG_B_). The CG did not carry out the training, but they did conduct the evaluations in the same period of the experimental groups (baseline, after the 16th and 32nd session).

The TG_A_ performed the IT protocol composing six series of 4 min at 55–60% HR_res_ with 1 min at 70–75% HR_res_, and the TG_B_ performed the same number of series, but instead with 4 min at 45–50% HR_res_ and with 1 min at 60–65% HR_res_. Therefore, the TG_B_ carried out interventions with lower intensities. Our findings demonstrated that both training groups achieved similar results, with no significant differences (*p* > 0.05). In the same sense, intra-group results did not show significant differences (*p* > 0.05) for all evaluated variables. However, with ES, it was possible to demonstrate the magnitude of chronic responses in HRR, BP, HRV and VO_2max_ (mL·kg^−1^·min^−1^), resulting from the intervention with interval training with the same protocol but with different intensities. With the application of analysis ∆%, it was also possible to perceive the magnitude of results obtained in the investigated variables when comparing evaluations after the 16th and 32nd sessions with the baseline.

The present study organized the analyses and interventions similarly to what Pichot et al. [18] applied. However, in this research, the authors performed the intervention with only one group for 14 weeks, four sessions per week, and the protocol was performed on a cycle ergometer containing nine series of 4 min at 65% HR_max_ with 1 min at 85% HR_max_, totaling a 45 min volume. In the present study, the interventions were carried out on a treadmill, accounting for 32 sessions distributed in 3 weekly sessions. The recovery time (4 min) and stimulus time (1 min) was the same as that applied by Pichot et al. [18]. However, the intervention comprised six sets, totaling 30 min of training, in three groups, two trainings (TG_A_ and TG_B_) and one control (CG).

Regarding autonomic analysis, IT is an efficient method for promoting improvements. In the hemodynamic analysis, studies that intervened with IT observed positive results. Pichot et al. [18] demonstrated positive results in HRR, SBP, DBP and MBP (*p* < 0.05). Molmen et al. [36] applied IT to active and sedentary older people and observed improvements in HRR, SBP and DBP (*p* < 0.05). Nemoto et al. [60] improved women’s HRR, SBP and DBP after five weeks of intervention with IT using, as in the present study, walking as an activity, demonstrating that this type of exercise in the elderly is a positive strategy. In addition to Pichot et al. [18], other experiments also obtained positive responses in HRV (*p* < 0.05) after the intervention with IT [25,28]. For a cardiorespiratory assessment, the IT was shown to have potential in the results, significantly improving (*p* < 0.05) VO_2max_ [18,35,36,37]. These findings are essential since the elderly reduce approximately 5% and 10% of cardiorespiratory capacity for active and sedentary individuals, respectively [47]. Considering the magnitudes of the results obtained, visualized by ES and by the ∆%, our findings corroborate those found by the studies mentioned earlier because we found no statistically significant difference in the crude analyses (*p* > 0.05).

The intensity of the stimuli still seems to be an unresolved issue because studies have shown positive responses with work intensities below thresholds [28] and with high-intensity stimuli above metabolic and/or ventilatory limits [36]. In the present study, the IT intervention was performed in two groups (TG_A_ and TG_B_) with different stimulus intensities of 70–75% HR_res_ (TG_A_) and 60–65% HR_res_ (TG_B_), and even with light to moderate stimulus the elderly people investigated showed improved cardioprotective capacity through the analysis of ES and ∆%. With that, we can deduce that there is no need to subject older people to high intensities to achieve positive results, thus preventing these individuals from possible cardiovascular overload and osteoarticular injury.

None of the studies mentioned used the ES and ∆% analysis methods. Our findings with these applications were positive, showing significant ES for hemodynamic, autonomic and cardiorespiratory variables. Through the ∆%, it was also possible to demonstrate in percentage values relevant differences in the variables evaluated after the 16th and 32nd sessions for both training groups. These methods are valid and clinically reinforce how important IT can be for cardiovascular training for the elderly. The possible mechanisms for the responses obtained within the elderly are still uncertain but plausible. Some cardiovascular adaptations can be affected in the elderly, such as central and peripheral functions. The aerobic stimulus can reduce the plasma level of renin, reflecting in the decrease in the renin-angiotensin system, improving the baroreflex activity and, consequently, the behavior of BP, HR and HRV [18,28]. Improving capillary density, endothelial function, and tissue oxygen delivery can influence hemodynamic and autonomic development [16,37,60]. Concerning VO_2max_, successive stimuli combined with recovery periods make the elderly impose higher intensities, promoting better transport capacity and consumption of O_2_, in addition to improving plasma hemoglobin and myoglobin volumes in muscles and increasing muscle capacity [35].

### Limitations, Directions for Further Research and Practical Applications

The present study is has some limitations, which may have influenced the results obtained. Even within the value stipulated by the sample calculation, the number of participants could have been larger for each group, but the difficulty of finding healthy elderly people available for the research study hindered this purpose. We could have had two groups (one training and one control), because we would have had the appropriate sample according to the calculation, but our goal was to generate comparisons between different training groups (different intensities), plus control, to demonstrate the potential of each intensity level and thus provide possible information for prescription with this population. Therefore, we believe that with a larger sample in each group, statistically we could have more significant results. On the other hand, we were able to show that the findings, in part, were positive through effect size and percentage delta. The sample in this study was classified as overweight by the body mass index (26.8 ± 1.3 kg/m**^2^**). It may have influenced the results because body composition directly affects cardiovascular and cardiorespiratory behavior.

Regarding the applied analysis, studies used equipment with a high level of reliability, such as for hemodynamic and autonomic evaluations performed with electrocardiogram and gas analyzer to verify VO_2max_ in maximum tests [18,35,36]. Our study was carried out with equipment with better accessibility with high and validated power reliability. We chose to check VO_2max_ in an estimated way. However, studies using the estimated cardiorespiratory analysis method achieved positive results [7], which allows us to accept that the estimated method can be a proper way of evaluating this functionality even with limitations. Another issue that may have influenced the results with low expressiveness (*p* > 0.05) is the intensity of stimulus applied, both for TG_A_ (70–75% HR_max_) and TG_B_ (60–65% HR_max_). However, we proposed a work with intensities that, in a certain way, are accessible to most of the elderly, making it possible for them to sustain the exercise. Another bias that limited our results was the time of each stimulus (1 min), which possibly, along with the intensity, was insufficient to promote physiological adaptations relevant to the greater hemodynamic, autonomic and cardiorespiratory responses. In addition, our study conducted the interval by a 1:4 ratio (1 min stimulus × 4 min recovery), thus enabling greater recovery after each stimulus which may have facilitated lower physiological demands. On the other hand, a longer stimulus time or higher intensity or higher density (lower ratio between stimulus and recovery), could promote greater fatigue and/or limitation (peripheral and/or central) in the participants and thus, they would not be able to complete the session or even the whole training period. Finally, the paucity of studies that have intervened with IT not considered high-intensity IT (HIIT), especially in the elderly, has limited our discussions of our findings.

We suggest further studies with the elderly using IT as an intervention on cardioprotective variables. Chronic studies using different intensities and protocols on hemodynamic, autonomic and cardiorespiratory behavior. We believe that larger stimuli are positive in these mentioned responses. With these possibilities, we will have more information about IT in the elderly population. However, our study has high potential in practical applicability because we applied protocols that are accessible to a large part of the elderly population. Other issues are that we used reliable assessments that were also very accessible and our intensity monitoring (HR) protocol, besides being accessible, was easy to understand and use.

## 5. Conclusions

With the present experiment results, it is possible to accept that the variables evaluated here are interdependent since the effects caused by IT in both training groups were similar. This reinforces the hypothesis of the high integration of the hemodynamic, autonomic and cardiorespiratory systems. In elderly individuals, responses resulting from different exercises are more discrete, as there is a possible resistance of the entire system because of the decrease in physiological efficiency due to the aging process. However, IT (using walking as an exercise) can be an essential strategy in training prescription for conditional improvement of cardioprotective function variables in physically active and healthy older people. However, other studies are suggested to establish the efficiency of this method for cardiovascular and cardiorespiratory variables in the elderly.

## Figures and Tables

**Figure 1 ijerph-20-05619-f001:**
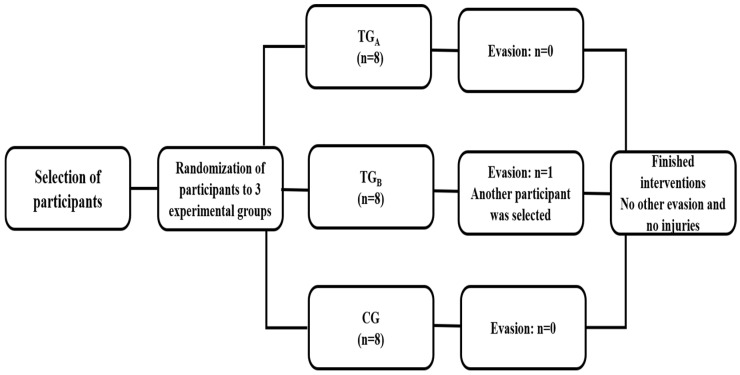
Flowchart of the entire intervention process for all groups.

**Table 1 ijerph-20-05619-t001:** Anthropometric characteristics and baseline variables of the participants.

Variables	TG_A_	TG_B_	CG
	M ± SD	M ± SD	M ± SD
Age (years)	65.1 ± 4.3	73.1 ± 7.2	68.2 ± 6.6
Weight (kg)	81.9 ± 13.1	74.2 ± 7.3	75.8 ± 5.2
Height (m)	1.71 ± 0.06	1.69 ± 0.06	1.71 ± 0.04
BMI (kg/m^2^)	27.8 ± 1.4	25.8 ± 1.2	26.8 ± 1.4
HR_R_ (bpm)	73 ± 11	76 ± 14	78 ± 7
SBP (mm/Hg)	129 ± 5	128 ± 10	126 ± 9
DBP (mm/Hg)	80 ± 3	77 ± 6	80 ± 4

**Table 2 ijerph-20-05619-t002:** Hemodynamic and cardiorespiratory variables with *p* values, effect size and Δ% for each condition.

	TG_A_			TG_B_			CG		
	*p*-Value	ES	∆%	*p*-Value	ES	∆%	*p*-Value	ES	∆%
**HR_R_**									
**Post 16°**	*p* > 0.099	−0.06 (Small)	−2%	*p* = 0.999	0.14 (Small)	3%	*p* = 0.999	0.13 (Small)	−2%
**Post 32°**	*p* = 0.846	−0.31 (Small)	−5%	*p* = 0.943	−0.31 (Small)	−6%	*p* > 0.999	0.14 (Small)	−2%
**SBP**									
**Post 16°**	*p* = 0.316	−0.86 (Large)	4%	*p* = 0.851	0.67 (Medium)	−5%	*p* = 0.639	0.51 (Medium)	4%
**Post 32°**	*p* = 0.148	−1.11 (Large)	6%	*p* = 0.991	−0.29 (Small)	−2%	*p* > 0.999	0.08 (Small)	0.1%
**DBP**									
**Post 16°**	*p* > 0.999	0.12 (Small)	0.1%	*p* > 0.999	−0.04 (Small)	−0.1%	*p* = 0.999	0.23 (Small)	1%
**Post 32°**	*p* = 0.999	−0.20 (Small)	−0.1%	*p* > 0.999	0.00 (Small)	0.1%	*p* > 0.999	0.08 (Small)	−0.1%
**MBP**									
**Post 16°**	*p* = 0.416	0.75 (Medium)	−3%	*p* = 0.860	−0.71 (Medium)	−4%	*p* = 0.821	0.51 (Medium)	3%
**Post 32°**	*p* = 0.103	−1.10 (Large)	−4%	*p* = 0.997	−0.29 (Small)	−2%	*p* > 0.999	0.09 (Small)	0.1%
**DP**									
**Post 16°**	*p* = 0.913	−0.27 (Small)	−5%	*p* = 0.961	−0.15 (Trivial)	−3%	*p* = 0.905	0.12 (Trivial)	2%
**Post 32°**	*p* = 0.733	−0.52 (Medium)	−10%	*p* = 0.733	−0.44 (Small)	−9%	*p* = 0.942	−0.09 (Trivial)	−1%
**VO_2max_**									
**Post 16°**	*p* = 0.928	0.82 (Large)	11%	*p* > 0.999	0.31 (Small)	5%	*p* = 0.998	0.00 (Small)	0.001%
**Post 32°**	*p* = 0.279	1.12 (Large)	15%	*p* = 0.804	0.38 (Small)	6%	*p* > 0.999	0.03 (Small)	0.07%

**TG_A_** = Training Group A; **TG_B_** = Training Group B; **CG** = control group; **Post 16°** = sixteen weeks post intervention; **Post 32°** = thirty-two weeks post intervention; **HR_R_** = resting heart rate; **SBP** = systolic blood pressure; **DBP** = diastolic blood pressure; **MBP** = mean blood pressure; **DP** = double product; **VO_2max_** = maximal oxygen consumption.

**Table 3 ijerph-20-05619-t003:** HRV indices with *p*-values, effect size and Δ% for each condition.

	TG_A_			TG_B_			CG		
	*p*-Value	ES	∆%	*p*-Value	ES	∆%	*p*-Value	ES	∆%
**RR**									
Post 16°	*p* = 0.999	0.14 (Small)	2%	*p* = 0.991	0.26 (Small)	5%	*p* = 0.986	0.46 (Small)	3%
Post 32°	*p* = 0.116	1.36 (Large)	20%	*p* = 0.765	0.42 (Small)	8%	*p* = 0.919	0.59 (Medium)	4%
**RMSSD**									
Post 16°	*p* = 0.999	−0.20 (Small)	−12%	*p* > 0.999	−0.01 (Small)	−2%	*p* = 0.999	0.17 (Small)	10%
Post 32°	*p* > 0.999	0.12 (Small)	11%	*p* > 0.999	0.12 (Small)	14%	*p* > 0.999	0.10 (Small)	6%
**SDNN**									
Post 16°	*p* > 0.999	−0.06 (Small)	−2%	*p* = 0.646	0.34 (Small)	25%	*p* = 0.999	0.33 (Small)	13%
Post 32°	*p* = 0.607	−0.72 (Medium)	−24%	*p* = 0.219	0.75 (Medium)	54%	*p* = 0.999	0.34 (Small)	14%
**LF**									
Post 16°	*p* = 0.999	−0.13 (Small)	−5%	*p* = 0.698	−1.16 (Large)	−9%	*p* = 0.805	0.54 (Medium)	14%
Post 32°	*p* = 0.556	−0.49 (Medium)	−19%	*p* = 0.349	−4.00 (Large)	−32%	*p* = 0.981	0.30 (Small)	7%
**HF**									
Post 16°	*p* = 0.999	0.13 (Small)	14%	*p* = 0.605	0.98 (Large)	40%	*p* = 0.965	−0.34 (Small)	−27%
Post 32°	*p* = 0.139	0.50 (Medium)	52%	*p* = 0.540	1.53 (Large)	62%	*p* = 0.993	−0.27 (Small)	−21%
**LF/HF**									
Post 16°	*p* = 0.997	−0.32 (Small)	−24%	*p* = 0.983	−0.58 (Medium)	−20%	*p* = 0.888	1.16 (Large)	91%
Post 32°	*p* = 0.191	−0.90 (Large)	−68%	*p* = 0.920	−0.80 (Large)	−28%	*p* = 0.999	0.25 (Small)	20%

**TG_A_** = Training Group A; **TG_B_** = Training Group B; **CG** = control group; **Post 16°** = sixteen weeks post intervention; **Post 32°** = thirty-two weeks post intervention; **RR** = time between two adjacent heart beats; **RMSSD** = square root of the mean of the square of the successive differences between the adjacent normal RR intervals; **SDNN** = standard deviation of all the normal RR intervals recorded in a time interval; **LF** = low frequency; **HF** = high frequency; **LF/HF** = low frequency/high frequency ratio.

## Data Availability

The datasets analyzed during the current study are available from the first author (Sant’Ana, L.) upon immediate request.

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
