# Peer review of "Chronic Effects of Different Intensities of Interval Training on Hemodynamic, Autonomic and Cardiorespiratory Variables of Physically Active Elderly People"

_ijerph, 2023, doi:10.3390/ijerph20095619_

Round 1
Reviewer 1 Report
General comments
The aim of the study was to verify the chronic effects of IT with different intensities on hemodynamic, autonomic, and cardiorespiratory variables of physically active older people. Participants were grouped into three experimental groups: training group A (TGA, n = 8), training group B (TGB, 24 n = 8) and control group (CG, n = 8). The authors concluded that IT can be a strategy in prescribing training for conditional improvement of cardioprotective variables in physically active and healthy elderly.
Introduction
Lines 59-60: "[19] mention that aging is not a limiting factor ..." - sentence is unclear.
Line 141-142: "The evaluations were performed in the pre (baseline) moments after the 16th and 32nd intervention sessions." Do authors mean there were three assessments overall, namely baseline, 6th session and 32nd session? It's not too clear.
Materials and Methods:
Lines 108-109: "... require a total sample size of 21 individuals" is repeated twice.
Line 129: Don't the participants need to consent to study before completing PAR-Q?
Results
Tables 2 & 3: Is "EF" same as "ES"? It's confusing reading about ES in the Results section but not finding it in the table.
The authors classified ES as small, medium, and large (line 229). However, there is a fourth classification - trivial - in Tables 2 & 3. Could the authors kindly explain? Is "moderate in line 245 equivalent to the "medium" used in tables 2 & 3?
Discussion:
Line 365: "TI" should be "IT".
The authors mentioned the 1-minute stimulus time was insufficient (lines 355-357). How did they choose it in the first place? Was it based on previous studies done?
Author Response
Dear Reviewer,
Thank you very much for all your input on our study! All the requested adjustments were made, and we confess that our study has become even richer.
Thank you very much!
Here are our arguments about your review:
“The aim of the study was to verify the chronic effects of IT with different intensities on hemodynamic, autonomic, and cardiorespiratory variables of physically active older people. Participants were grouped into three experimental groups: training group A (TGA, n = 8), training group B (TGB, 24 n = 8) and control group (CG, n = 8). The authors concluded that IT can be a strategy in prescribing training for conditional improvement of cardioprotective variables in physically active and healthy elderly”
R= It has been corrected! We used the word "cardioprotection" because of the word restriction in the abstract. This is because the variables we evaluated are responsible for a cardioprotective action.
Introduction
Lines 59-60: "[19] mention that aging is not a limiting factor ..." - sentence is unclear.
R= It has been corrected, we have rewritten the explanation.
Line 141-142: "The evaluations were performed in the pre (baseline) moments after the 16th and 32nd intervention sessions." Do authors mean there were three assessments overall, namely baseline, 6th session and 32nd session? It's not too clear.
R= We agree with the placement and have improved the explanation about the moments of analysis.
Materials and Methods:
Lines 108-109: "... require a total sample size of 21 individuals" is repeated twice.
R= It has been corrected.
Line 129: “Don't the participants need to consent to study before completing PAR-Q?”
R= It has been corrected.
Results
“Tables 2 & 3: Is "EF" same as "ES"? It's confusing reading about ES in the Results section but not finding it in the table.”
R= Yes, the correct is ES. There was a typo. It has been corrected.
“The authors classified ES as small, medium, and large (line 229). However, there is a fourth classification - trivial - in Tables 2 & 3. Could the authors kindly explain? Is "moderate in line 245 equivalent to the "medium" used in tables 2 & 3?”
R= It has been corrected. Actually "trivial" was put in when the value was considered insignificant. We put all of them as also "small".
Discussion:
Line 365: "TI" should be "IT".
R= It has been corrected
“The authors mentioned the 1-minute stimulus time was insufficient (lines 355-357). How did they choose it in the first place? Was it based on previous studies done?”
R= We explain our argument further. The protocol was based on the study by Pichot et al. (2005), cited in the present study.
Please see the attachment.

Reviewer 2 Report
ABSTRACT
In the abstract, authors should report a brief background and the aim of the study before the methods.
Also in the abstract, authors should define the meaning of “IT” the first time it appears.
INTRODUCTION
(line 85-89): “However, conditional improvements in this population … minimizing the deleterious effects caused by the low physical fitness added to the aging process”. Among these physical characteristics, the body balance is of fundamental importance for the prevention of the risk of falls.
METHODS
(line 105): Please report the sample size power analysis description in the statistical analysis section.
Please clarify the sampling method.
Please report the randomisation procedure.
More details on training protocols are needed. Setting? Trainer? Type of exercises?
Considering the study design, why authors did not compute a mixed ANOVA with repeated measures using a a between- and within-subject factor with factor “group” as between-subject and factor “time” as within-subject?
RESULTS
Based on my last comment, results should be rearranged.
Author Response
Dear Reviewer,
Thank you very much for all your input on our study! All the requested adjustments were made, and we confess that our study has become even richer.
Thank you very much!
Here are our arguments about your review:
ABSTRACT
“In the abstract, authors should report a brief background and the aim of the study before the methods”
R= We include this information.
“Also in the abstract, authors should define the meaning of “IT” the first time it appears.”
R= It has been corrected
INTRODUCTION
(line 85-89): “However, conditional improvements in this population … minimizing the deleterious effects caused by the low physical fitness added to the aging process”. “Among these physical characteristics, the body balance is of fundamental importance for the prevention of the risk of falls”
R= We include this information.
METHODS
(line 105): “Please report the sample size power analysis description in the statistical analysis section”.
R= It has been corrected
“Please report the randomisation procedure”
R= We include this information.
“More details on training protocols are needed. Setting? Trainer? Type of exercises?”
R= We enter all this information.
“Considering the study design, why authors did not compute a mixed ANOVA with repeated measures using a a between- and within-subject factor with factor “group” as between-subject and factor “time” as within-subject?”
R= We redo the statistics and adjust the results. It was still not significant. We did not pay attention to the mixed model issue. We corrected it, because our groups were independent.
RESULTS
“Based on my last comment, results should be rearranged.”
R= It has been corrected
Please see the attachment

Round 2
Reviewer 2 Report
The authors solved almost all of my previous comments. However, the authors should clarify the randomization process used. I am attaching a link for reference: "Kang M, Ragan BG, Park JH. Issues in outcomes research: an overview of randomization techniques for clinical trials. J Athl Train. 2008 Apr-Jun;43(2):215-21".
Also, since the authors argue walking and improving body balance for the prevention of falls (lines 94-95), I suggest inserting the following reference: "Battaglia G, et al. Walking in Natural Environments as Geriatrician’s Recommendation for Fall Prevention: Preliminary Outcomes from the “Passiata Day” Model. Sustainability. 2020; 12(7):2684".
Author Response
Dear reviewer,
Thank you very much for your contributions.
About the randomization, we explain in full detail how it was performed. And we took the opportunity to put the suggested reference. Normally, we were not used to explain the whole process, but we realized that this enriched even more the writing of our methodology.
And about, the suggested reference we included in the paragraph about the importance of balance in minimizing the risk of falls.
We thank you for all your contributions.